# Surviving the Storm: The Impact of COVID-19 on Cervical Cancer Screening in Low- and Middle-Income Countries

**DOI:** 10.3390/healthcare11233079

**Published:** 2023-12-01

**Authors:** Mandana Vahabi, Anam Shahil-Feroz, Aisha Lofters, Josephine Pui-Hing Wong, Vijayshree Prakash, Sharmila Pimple, Kavita Anand, Gauravi Mishra

**Affiliations:** 1Daphne Cockwell School of Nursing, Toronto Metropolitan University, 350 Victoria Street, Toronto, ON M5B 2K3, Canada; 2Peter Gilgan Centre for Women’s Cancers, Women’s College Hospital Research Institute, Toronto, ON M5G 1N8, Canada; 3Institute of Health Policy Management and Evaluation, Dalla Lana School of Public Health, University of Toronto, Toronto, ON M5T 3M6, Canada; 4Department of Family & Community Medicine, University of Toronto, St. Catharines, ON M5G 1V7, Canada; 5Dalla Lana School of Public Health (Cross-Appointed), University of Toronto, Toronto, ON M5T 3M7, Canada; 6Department of Preventive Oncology, Centre for Cancer Epidemiology, Tata Memorial Centre, Homi Bhabha National Institute (HBNI), R. No. 314, 3rd Floor, Service Block, E Borges Marg, Mumbai 400012, India

**Keywords:** COVID-19, cervical cancer screening, HPV self-sampling, Pap test, LMICs, India

## Abstract

According to the Center for Disease Control and Prevention’s National Breast and Cervical Cancer Early Detection Program, the cervical cancer screening rate dropped by 84% soon after the declaration of the COVID-19 pandemic. The challenges facing cervical cancer screening were largely attributed to the required in-person nature of the screening process and the measures implemented to control the spread of the virus. While the impact of the COVID-19 pandemic on cancer screening is well-documented in high-income countries, less is known about the low- and middle-income countries that bear 90% of the global burden of cervical cancer deaths. In this paper, we aim to offer a comprehensive view of the impact of COVID-19 on cervical cancer screening in LMICs. Using our study, “Prevention of Cervical Cancer in India through Self-Sampling” (PCCIS), as a case example, we present the challenges COVID-19 has exerted on patients, healthcare practitioners, and health systems, as well as potential opportunities to mitigate these challenges.

## 1. Introduction

Cervical cancer, caused primarily by Human Papillomavirus (HPV), is a significant health issue globally, with around 90% of related deaths occurring in low- and middle-income countries (LMICs) [1,2]. In particular, India faces a substantial burden, accounting for approximately 20% of worldwide cases, with cervical cancer being the second leading cause of cancer-related deaths among Indian women, including those in their prime years [3]. While cervical cancer is highly preventable through early detection and treatment, LMICs struggle to adopt preventive measures due to limited healthcare infrastructure, sociocultural factors (modesty, male dominance in making decisions about family matters, etc.), social stigma surrounding sexually transmitted infections, limited knowledge about cervical cancer and screening, and a lack of female healthcare providers [4,5,6,7]. As a result, screening coverage in LMICs is only 19%, compared to 63% in high-income countries [8].

Although LMICs were making efforts to advance their cervical cancer screening, the profound impact of the COVID-19 pandemic has disrupted global cancer screening efforts. The strain on healthcare systems caused by the pandemic is expected to disproportionately impact LMICs with limited resources. While the impact of the COVID-19 pandemic on cancer screening in high-income countries is well-documented [9,10,11,12,13,14], less is known about LMICs primarily due to insufficient research and documentation. Based on the limited evidence available, LMICs such as Bangladesh, Morocco, and Argentina also experienced a substantial reduction in cancer screening uptake [15]. 

Our project, “Prevention of Cervical Cancer in India through Self-Sampling” (PCCIS) [14], which was carried out during COVID-19, aimed to increase the uptake of cervical cancer screening among under- or never-screened (UNS) Indian women in rural areas through HPV self-sampling (HPV-SS) and family-centered, arts-based sexual health literacy intervention. Like other LMICs, we faced several challenges including the need to adhere to COVID-19 protocols such as social distancing, staffing issues stemming from re-deployment difficulties, and the prioritization of COVID-19 issues over cervical screening. 

In this paper, we aim to explore whether similar challenges were encountered in promoting cervical cancer screening in other LMICs during the pandemic. We present the challenges COVID-19 has exerted on patients, healthcare practitioners, and health systems, as well as potential opportunities that could help address these challenges. The exploration is grounded in extensive discussions with the project team during implementation, aimed at adapting the study components to align with COVID-19 pandemic control measures, and informed by a review of the literature. 

Despite the efforts of the National Programme for Prevention and Control of Cardiovascular Disease, Diabetes, Cancer, and Stroke (NPCDCS) since 2016, cervical cancer screening in India remains primarily opportunistic. The program faced significant challenges during the COVID-19 waves, leading to the suspension of non-COVID-19 medical services, including cancer screening, from May 2020 onwards [15,16,17,18]. While this opinion piece is rooted in the Indian context, focusing on the challenges encountered during the PCCIS project, the identified challenges and opportunities discussed herein hold broader relevance for other low- and middle-income countries (LMICs) that lack organized screening programs.

## 2. Challenges Imposed by COVID-19 in the Context of Cervical Cancer Screening for Patients, Healthcare Practitioners, and Health Systems in LMICs

According to the Center for Disease Control and Prevention’s National Breast and Cervical Cancer Early Detection Program, cervical cancer screening significantly dropped by 84% soon after the COVID-19 pandemic was declared [19]. This drop was accompanied by a sharp decline in online search interest in cervical cancer and the Pap test after the pandemic was declared [20]. The challenges faced for cervical cancer screening were largely attributed to the required in-person nature of the screening process [12] and the measures implemented to control the spread of the virus. In line with the approach used by Cancino et al. [21], we organized the evidence and insights related to the challenges presented by COVID-19 to cervical cancer screening with a specific focus on three key stakeholders: patients, healthcare practitioners, and health systems.

### 2.1. Patients

The pandemic had a substantial impact on cervical cancer screening for patients, primarily due to movement restrictions and limited access to healthcare facilities during lockdowns [11,22]. Mitigation measures like social distancing and travel restrictions, coupled with the closure of non-emergency health services, led to substantial disruptions in cancer care, particularly in screening services, resulting in delayed access to vital healthcare [11]. The impact of these restrictions was evident in Bangladesh, where evidence indicates a significant decrease in the number of screening tests conducted during the pandemic [22]. In a related study, Lucas et al. examined the impact of the COVID-19 pandemic on cancer screening programs, identifying a notable drop in screening volumes due to people’s fear of infection [11,23]. In a cross-sectional study conducted across 13 LMICs, patients rated the availability of medical services on a scale of 0 to 100, with over 60% giving a score of 50 or lower for cancer screening services [24]. A majority expressed strong concerns about noncompliance and staff overload affecting screening programs [24]. In our study, the research team faced additional challenges, including power outages, water scarcity, and transportation strikes, during data collection, which involved face-to-face questionnaires and focus groups. Our study populations in Shirgaon, Jamsar, and Khodala had poor knowledge of cervical cancer screening. Consequently, these challenges were not perceived as barriers to accessing screening but rather as issues our team had to navigate for infection control during education sessions. In addition, work-related interference emerged as a notable factor hindering prevention efforts in our study. Male participants faced challenges attending SHE sessions because of their daytime occupations, which included jobs in fisheries, farming, or nearby urban areas. To accommodate, one-on-one sessions for men were scheduled in the evening or on weekends. 

### 2.2. Healthcare Providers

The pandemic also had a significant impact on healthcare providers involved in cervical cancer screening. Most providers in LMICs relied on pelvic examinations for cervical cancer screening, which involve close contact and extended time in enclosed spaces [12]. These factors made it challenging for healthcare providers to maintain appropriate social distancing and adhere to strict infection prevention protocols. Additionally, the need for thorough disinfection procedures between patients not only extended waiting times but also significantly reduced the overall service capacity that healthcare providers could accommodate [19]. Many providers had to undergo additional training to adapt to new safety protocols and incorporate telehealth solutions for consultations and follow-ups [25]. Moreover, some LMICs experienced workforce shortages as healthcare professionals were redirected to address COVID-19-related needs, further reducing their capacity for cervical cancer screening. These challenges further strained the already stretched resources of healthcare facilities and impacted the ability of healthcare providers to offer timely and comprehensive cervical cancer screening services [12].

### 2.3. Health Systems

The COVID-19 pandemic brought significant disruptions to healthcare systems and routine medical services, impacting programs such as cervical cancer screening and HPV vaccinations. The COVID-19 outbreak forced most health systems to halt or slow down ‘non-urgent’ services in order to reassign staff for COVID-19-related work and reduce the influx of patients to these health facilities [11]. Cervical cancer screening was included among such services deemed ‘non-urgent’ in most countries. The consequent reduction in the volume of cancer screening tests resulted in a substantial rise in avoidable cancer deaths [26]. For example, in El Salvador’s Occidental and Oriental regions, thousands of HPV-positive women faced delays in their follow-up care [27]. A collaborative effort with external funding engaged private colposcopists and pathology laboratories to locate and provide care for these women. High rates of high-grade pre-cancer (23%) and invasive cancer (1.3%) were detected. The HPV vaccination program also faced challenges due to school closures, achieving only 28% of its target in 2020 [27]. 

In India, the pandemic had a profound impact on the Indian healthcare system, particularly disrupting the infrastructure for cancer screening. This disruption occurred as healthcare resources were redirected, leading to the temporary halt of screening services. Additionally, travel restrictions during lockdowns hindered access to health services in metro cities, causing delays in early screening, accurate diagnoses, and timely treatments. Consequently, this delay potentially led to undetected cases and an increased burden of advanced-stage cancers [12,16,28]. The missed screenings not only affect future cancer mortality rates but also result in fewer detections of early-stage cancer and pre-cancerous conditions due to the absence of follow-up assessments. Our study encountered unique health systems challenges, including closed lodging facilities, water shortages, the need for additional staff training, the restriction of project execution to non-contamination areas, and transportation difficulties. This led the research team to share a government guesthouse, rely on the parent organization for transportation to and from villages, and walk to campsites with the study materials. These challenges add significant strain to the already overburdened healthcare systems in LMICs, potentially leading to more advanced and costly cancer cases, thereby intensifying the challenges faced by these resource-limited systems.

## 3. Opportunities and Solutions

The challenges faced in cervical cancer screening during the pandemic highlighted the need for strategies to mitigate the impact and ensure access to essential healthcare services. Our project employed several approaches to address COVID-19 challenges, some of which are supported by international studies [12,29,30,31,32], including: adopting primary HPV screening with self-sampling; involving family members, particularly male partners, in sexual health education, as recommended by WHO [32]; empowering community health workers for an improved outreach; and building local capacity through virtual training and the use of art-based information/education materials in local languages. These strategies have shown promise in improving knowledge and attitudes, increasing screening access, reducing patient–provider contact, prioritizing individuals at higher risk of cervical cancer, and helping to ensure study completion despite a global pandemic. For instance, our study observed advancements in participants’ knowledge and attitudes about cervical cancer and screening, coupled with a reduction in STI stigma following their engagement in SHE sessions (overall mean difference in knowledge: z = 6.1 ± 2.4, *p* < 0.001; attitudes towards the Pap test: z = 2.2 ± 8.4, *p* < 0.001; STI stigma: z = 2.8 ± 12.4, *p* < 0.001). Among the 120 women enrolled in our study, 115 chose HPV-SS as the primary screening test, while 2 opted for Pap and 1 chose VIA testing, demonstrating an increase in screening participation [7]. 

### 3.1. Adoption of Primary HPV Screening with Self-Sampling

The adoption of HPV self-sampling (HPV-SS) has shown promise in increasing screening participation and building resilience within cervical cancer screening systems against future disruptions in care, and was one of the key interventions used in our study [29]. Gorin et al.’s research highlights the accessibility and acceptability of cervical self-screening among patients, suggesting home-based cancer screening as a viable post-pandemic option alongside telemedicine. Similarly, our study showed a high uptake of cervical cancer screening through the use of HPV-SS. As described in our published study results, almost all of the female participants in our study opted for HPV-SS (i.e., 115/120) and all participants (male and female) showed improvements in their knowledge and attitudes towards cervical cancer and screening, and a reduction in the stigma surrounding sexually transmitted diseases (STIs) [14]. HPV-SS appears more resilient to pandemic-related disruptions and can be seamlessly integrated into the workflows of primary care providers, enhancing the effectiveness of follow-up screening [30]. The experience gained from the COVID-19 pandemic can expedite the adoption of primary HPV screening with self-sampling in both clinical and community settings. The increased production and distribution of molecular tests and nucleic acid testing platforms, along with investments in laboratory and manufacturing capacity driven by the COVID-19 response, can improve access [12,31].

### 3.2. Developing Health System Capacity

Strengthening health system capacity is vital to ensuring efficient cervical cancer screening programs. This includes augmenting healthcare infrastructure, such as improving laboratory facilities for cytology-based screening and increasing access to HPV testing. To manage the timely reporting of HPV tests, our team in Mumbai designated and trained laboratory staff specifically for processing the HPV-SS samples received from participants, thus enabling smooth handling of the screening tests and seamless processing of the COVID-19 tests. Additionally, building a workforce of health professionals and paramedical workers with screening awareness through training and capacity-building initiatives is crucial for successful program implementation [17,31]. For example, the involvement of community health workers can significantly contribute to enhancing a community’s awareness of cervical cancer screening, streamlining the screening and follow-up processes, and providing essential education and counselling services. Furthermore, it can play a pivotal role in promoting the adoption of HPV self-sampling, which has emerged as a promising approach to increase screening participation and make screening more accessible, especially in underserved communities [31]. 

In our study, we relied heavily on six community health workers to recruit and educate women and their family members about sexual health, cervical screening, and HPV self-sampling. These workers were trusted members of the community who knew how to communicate with participants in a culturally appropriate manner. They were instrumental in the success of our study, where only 2 out of 120 women participants refused to be screened. Furthermore, we trained “Accredited Social Health Activists" (Commonly referred to as ASHA workers, ‘ASHA’ meaning hope) to deliver sexual health education regarding cervical cancer using the art-based educational materials developed by our project. ASHAs provide a new model of care adopted by the Indian government that relies on building capacity in local and remote areas by recruiting trusted community members and training them to promote awareness about health-related issues that face their respective communities for immunization, and maternal and child health. Through this project, we intended to leverage their skills for cancer prevention so that they could continue their collaboration with the local partners and continue their training and peer-to-peer education in other neighboring districts facing similar issues. However, we were hindered by local government policies when hiring ASHA workers. According to a government policy in India, ASHA workers are volunteers working on honorariums, and not full-time salaried employees. Hence, granting us permission to hire ASHA workers and pay them salaries was believed to be against their mandate and considered unfair to the rest of the ASHA workers not hired for this project. Hence, we used ASHA workers in close collaboration with medical social workers (MSWs), and assisted them in their community mapping, introduction to families, recruitment of participants, and taking the screen-positive participants to the diagnostic camp. 

Considering the potential role of ASHA workers in increasing cervical screening uptake, we trained 451 ASHA workers and block facilitators at the taluka (subdivision of the district) levels of Shirgoan, Mokhada, and Jawhar (three rural villages in the Palghar district of Maharashtra, India, with minimal to zero cervical cancer screening rates) to deliver sexual health education regarding cervical cancer using the art-based educational materials developed by our project. All the attendees in the training session were compensated for their time and travel as per government regulation. They were also provided with copies of the PCCIS sexual health educational materials to continue their outreach and direct women towards cervical screening. This effort served to support “WHO’s call for action to eliminate cervical cancer,” a component of the Non-Communicable Disease Program (NCD). Furthermore, this training session for female ASHA workers helped to build capacity and empowered local ASHA workers by leveraging their skills to include the promotion of cervical cancer screening.

### 3.3. Building Local Capacity through Virtual Training and Culturally Appropriate Messaging 

Virtual training of healthcare providers in LMICs was implemented, with local support for on-site instruction, for building local capacities for cervical cancer screening despite the restrictions and social distancing requirements during the pandemic [33]. In our study, the international team included members from both Canada and India. Our team members from both countries provided virtual training to the community workers and healthcare providers involved in this study within their community. Additionally, we received invaluable on-site support from the Tata Memorial Centre for training community health workers and providing them with culturally appropriate health education materials for reducing HPV stigma and promoting cervical cancer screening.

## 4. Strengths and Limitations

This opinion piece has both strengths and limitations. First, it addresses a significant gap in the existing literature by exploring the impact of COVID-19 on cervical cancer screening in low–middle-income countries. Most of the current literature predominantly focuses on the challenges faced by high-income countries in cervical cancer screening or highlights the barriers in LMICs related to general cancer screening, diagnoses, and management during the pandemic, with limited attention to cervical cancer screening. Another strength of this paper lies in its application of Cancino et al.’s framework [21] to organize evidence on the challenges posed by COVID-19 for cervical cancer screening, with a specific focus on three key stakeholders: patients, healthcare practitioners, and health systems. This framework provides readers with a comprehensive understanding of the magnitude of challenges in this field, enabling the development of context-specific strategies.

This study also has some limitations. While rooted in the Indian context and based on findings from our PCCIS project conducted in India, this opinion piece’s reflections and conclusions may not capture the nuances of all LMICs in Africa and Asia. Readers must exercise caution when interpreting the challenges and opportunities presented, recognizing the need for context-specific considerations. Additionally, this opinion piece primarily emphasizes opportunities and solutions within the context of the PCCIS project, potentially limiting readers’ awareness of alternative strategies, such as the use of telemedicine for cervical cancer screening. There exists a broader spectrum of approaches that can be considered for promoting cervical cancer screening amid the challenges posed by COVID-19.

## 5. Conclusions

The COVID-19 pandemic had a significant and disruptive impact on cervical cancer screening, affecting patients, healthcare providers, and health systems in LMICs. These disruptions highlight the urgent need for innovative solutions, particularly in the realm of cervical cancer prevention and screening. As we adapt to the evolving landscape of healthcare delivery, we must proactively embrace opportunities such as adopting primary HPV screening with self-testing, empowering community healthcare workers for outreach, developing healthcare systems’ capacities, and building the local capacity for information, education, and communication of cervical screening messages in local languages through virtual training and on-site support. These practical solutions, used successfully in our research study, offer potential ways to improve accessibility, efficiency, and inclusivity within cervical cancer screening programs. Together, these strategies hold the transformative potential to reshape the landscape of cervical cancer prevention in LMICs.

## Data Availability

Data are contained within the article.

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
