# Peer review of "Surviving the Storm: The Impact of COVID-19 on Cervical Cancer Screening in Low- and Middle-Income Countries"

_healthcare, 2023, doi:10.3390/healthcare11233079_

Round 1
Reviewer 1 Report
Comments and Suggestions for Authors
This paper is important but confusing. The generalizability of one program to all LMICs seems farfetched, especially given that not all LMICs will face the same restrictions.
This paper spends a lot of time rehashing the effects of COVID-19 on the healthcare system where that could be confined to one paragraph in the background.
There is continual mention of cervical cancer screening rates in high income counties which is largely irrelevant to anything more than a brief introduction comparison.
This is 4 pages of describing the very well documented impact of COVID-19 on healthcare personnel and the system. I could not succinctly tell you what this program described in this manuscript was about, and that is what the manuscript promises in the abstract.
If cervical screenings were already low in the first place, this paper ends up being conjecture.
I would rewrite this paper as a case study and leave the generalization and COVID-19 impact out of it.
Comments on the Quality of English Languageminor revisions.
Author Response
Title: Surviving the Storm: The Impact of COVID-19 on Cervical Cancer Screening in Low and Middle-Income Countries
|
Response to Reviewer 1 Comments
|
|||
|
1. Summary |
|
|
|
|
Thank you for dedicating your time to reviewing this manuscript. Below, you will find detailed responses along with corresponding revisions and corrections highlighted in the resubmitted files. Upon thorough examination, it seems that the paper was assessed as if it were an original article. We would like to emphasize that our submission is classified as an opinion piece.
|
|||
|
2. Questions for General Evaluation |
Reviewer’s Evaluation |
Response and Revisions |
|
|
Does the introduction provide sufficient background and include all relevant references? |
Yes/Can be improved/Must be improved/Not applicable |
|
|
|
Are all the cited references relevant to the research? |
Yes/Can be improved/Must be improved/Not applicable |
|
|
|
Is the research design appropriate? |
Yes/Can be improved/Must be improved/Not applicable |
|
|
|
Are the methods adequately described? |
Yes/Can be improved/Must be improved/Not applicable |
|
|
|
Are the results clearly presented? |
Yes/Can be improved/Must be improved/Not applicable |
|
|
|
Are the conclusions supported by the results? |
Yes/Can be improved/Must be improved/Not applicable |
|
|
|
3. Point-by-point response to Comments and Suggestions for Authors
Comment 1: This paper is important but confusing. The generalizability of one program to all LMICs seems farfetched, especially given that not all LMICs will face the same restrictions. Response: Thank you for your comment. It's important to clarify that the purpose of this paper is not to assert the generalizability of our program. Instead, our aim is to highlight the challenges that COVID-19 has posed for patients, healthcare practitioners, and health systems in promoting cervical cancer screening in LMICs during the pandemic. We also explore potential opportunities that could help address these challenges. The exploration is based on in-depth discussions with the project team during implementation of our program, and informed by a thorough review of the literature.
Comment 2: This paper spends a lot of time rehashing the effects of COVID-19 on the healthcare system where that could be confined to one paragraph in the background. Response: Since we used Cancino et al.’s approach, we structured the evidence and insights concerning the challenges posed by COVID-19 to cervical cancer screening, with a particular emphasis on three primary stakeholders: patients, healthcare practitioners, and health systems. The objective was to elucidate how the COVID-19 pandemic significantly and disruptively influenced cervical cancer screening, impacting patients, healthcare providers, and health systems.
Comment 3: There is continual mention of cervical cancer screening rates in high income counties which is largely irrelevant to anything more than a brief introduction comparison. Response: We referenced cervical cancer screening rates in high-income countries to highlight that the impact of the COVID-19 pandemic on cancer screening is well-documented in these regions and there is limited information about the situation in LMICs, primarily due to inadequate research and documentation.
Comment 4: This is 4 pages of describing the very well documented impact of COVID-19 on healthcare personnel and the system. I could not succinctly tell you what this program described in this manuscript was about, and that is what the manuscript promises in the abstract. Response: We want to emphasize that this paper serves as an opinion piece, aiming to provide a comprehensive perspective on the impact of COVID-19 on cervical cancer screening in LMICs. Utilizing our study, 'Prevention of Cervical Cancer in India through Self-Sampling' (PCCIS), as a case example, we shed light on the challenges faced by patients, healthcare practitioners, and health systems due to COVID-19, along with potential opportunities to address these challenges. The original paper for this study has been published at: https://bmcpublichealth.biomedcentral.com/articles/10.1186/s12889-023-15602-1
Comment 5: If cervical screenings were already low in the first place, this paper ends up being conjecture. Response: As mentioned in the paper, we want to reiterate that LMICs were actively working to enhance cervical cancer screening under initiatives such as the National Programme for Prevention and Control of Cardiovascular Disease, Diabetes, Cancer, and Stroke (NPCDCS). However, the profound impact of the COVID-19 pandemic disrupted global cancer screening efforts. The NPCDCS program faced significant challenges during Covid-19 waves, leading to the suspension of non-Covid medical services, including cancer screening.
Comment 6: I would rewrite this paper as a case study and leave the generalization and COVID-19 impact out of it. Response: As stated earlier, this paper falls under the category of an opinion piece, with generalization not being the primary objective. The original study has already been published, and the aim of this paper is to document the challenges imposed by COVID-19 on patients, healthcare practitioners, and health systems, alongside exploring potential opportunities to address these challenges |
|||
Reviewer 2 Report
Comments and Suggestions for Authors
See enclosed comments.

See enclosed comments.
Author Response
|
Response to Reviewer 2 Comments
|
|
|||
|
1. Summary |
|
|
|
|
|
Thank you very much for taking the time to review this manuscript. Please find the detailed responses below and the corresponding revisions/corrections highlighted/in track changes in the re-submitted files.
|
|
|||
|
2. Questions for General Evaluation |
Reviewer’s Evaluation |
Response and Revisions |
|
|
|
Does the introduction provide sufficient background and include all relevant references? |
Yes/Can be improved/Must be improved/Not applicable |
|
|
|
|
Are all the cited references relevant to the research? |
Yes/Can be improved/Must be improved/Not applicable |
|||
|
Is the research design appropriate? |
Yes/Can be improved/Must be improved/Not applicable |
|||
|
Are the methods adequately described? |
Yes/Can be improved/Must be improved/Not applicable |
|||
|
Are the results clearly presented? |
Yes/Can be improved/Must be improved/Not applicable |
|||
|
Are the conclusions supported by the results?
|
Yes/Can be improved/Must be improved/Not applicable |
|||
- Point-by-point response to Comments and Suggestions for Authors
General Comments to Authors: This Opinion piece fills a gap in the literature in reviewing cervical cancer HPV screening in low- and middle-income countries during the COVID pandemic with reference to the authors’ program of HPV self-sampling taking place in India. The submission is a summation and reflection, as opposed to a full analysis of their study, which the authors have undertaken elsewhere. The piece is well-written, but could benefit from a few conceptual additions and grammatical changes
Response: Thanks so much for appreciating our work. We have made track changes to our paper for conceptual additions and grammar changes.
- 2, Para. 3, line 61:
Please explain whether the exploration you are undertaking in this piece is based on discussions within
your project team, a review of literature, a review of policies, or issues highlighted during study-related
collaboration.
Response: The exploration is grounded in extensive discussions with the project team during implementation, aimed at adapting the study components to align with Covid-19 pandemic control measures, and informed by a review of the literature. Despite the National Programme for Prevention and Control of Cardiovascular Disease, Diabetes, Cancer, and Stroke (NPCDCS) since 2016, cervical cancer screening in India remains largely opportunistic. The national cancer screening program faced setbacks during the Covid-19 waves, as healthcare resources were redirected, leading to the suspension of non-Covid medical services, including cancer screening. Efforts to resume services were impeded by pandemic-related challenges, such as delays in obtaining permissions and logistical issues. Transportation strikes, travel restrictions, and community resistance further complicated the research team's access to target areas. Community resistance, driven by fear of infection, emphasized the crucial role of local ASHA workers in building trust.
- 2, Para. 5:
line 80: In the U.S., work-related (e.g., working at restaurants that stayed open) interference with taking
COVID precautions was a factor in prevention efforts. Any relevance of this type of interference to the
populations in yours or related studies?
Response: Thank you for the comment. We have mentioned that work-related interference emerged as a notable factor hindering prevention efforts in our study. Male participants faced challenges attending SHE sessions because of their daytime occupations, which included jobs in fisheries, farming, or nearby urban areas. To accommodate, one-on-one sessions for men were scheduled in the evening or on weekends.
line 87: In your study, was data collected through surveys, questionnaires, interviews, or focus groups?
Response: Thank you for the comment. In our study the research team faced additional challenges, including power outages, water scarcity, and transportation strikes, during data collection, which involved face-to-face questionnaires and focus groups.
- 3, Para. 2, line 123:
Form a new paragraph starting “In India, the pandemic …”.
Response:. Noted and revised as suggested
- 3, Para. 3, lines 140-3:
Add a few statistics to reinforce your points about study-related improvements.
Response: Thank you for the comment. We have now added the following paragraph to provide evidence for improvements in screening, knowledge and attitudes, and STI stigma.
“Our study observed advancements in participants' knowledge and attitudes about cervical cancer and screening, coupled with a reduction in STI stigma following their engagement in SHE sessions (overall mean difference in Knowledge z = 6.1 ± 2.4, P < 0.001; attitudes about Pap-test: z = 2.2 ± 8.4, P < 0.001; STI stigma: z = 2.8 ± 12.4, P < 0.001). Among the 120 women enrolled in our study, 115 chose HPV-SS as the primary screening test, while 2 opted for Pap and 1 chose VIA testing, demonstrating an increase in screening participation.”
- 4, Para. 2, line 177:
This is an especially long paragraph. Suggest introducing a new paragraph starting with “In our study, we
relied heavily …”.
Response:. Noted and revised as suggested
- 5, Para. 1, line 199:
line 199: Form a new paragraph starting “Considering the potential role …”.
line 201: the taluka level -> the taluka level (several words of explanation)
Response: Noted and revised as suggested and explained that Taluka is a subdivision of a district.
- 1, Title:
Low and Middle-Income -> Low- and Middle-Income
Response: Revised as suggested .
- 2, Para. 2:
line 53: which carried out during COVID-19 aimed -> awkward; please rephrase
line 54: under -> under
Response: Revised as suggested .
- 2, Para. 4, line 65:
LMIC -> LMICs
Response: Revised as suggested.
- 3, Para. 2:
line 114: please find an alternate word to “footfalls”
lines 115, 123, 126: add a single space before the citation lead parenthesis
line 129: This adds -> This lack adds
Response: Revised as suggested
- 4, Para. 1:
line 145: self-sampling (HPV-SS)has -> self-sampling (HPV-SS) has
line 154: and screening and reduction -> and screening, and reduction
Response: Revised as suggested
- 4, Para. 2:
line 163: ensure -> ensuring
line 167: participants -> participants,
line 168: smooth handling the screening tests in addition to seamless -> smooth handling of the
screening tests and seamless
line 174: they can -> it can
Response: Revised as suggested
- 4, Para. 2, line 197:
with the Medical Social Workers -> with Medical Social Workers
Response: Revised as suggested
- 5, Para. 1:
line 203: deliver -> to deliver
line 207: This served -> This effort served
quotation mark (“) after the word “action” seems misplaced
line 208: cancer” a component -> cancer,” a component
Response Revised as suggested
- 5, Para. 2:
line 216: In our study, our international team -> In our study, the international team
line 219: Community Health workers -> community health workers
Response: Revised as suggested
- 5, Para. 3, line 228: Community Health Workers -> community health workers
Response: Revised as suggested
- 5, Para. 4, line 246:
Bold “Informed Consent Statement:”
Response: Revised as suggested
- 6, Para. 1, line 248:
Remove the indentation from the Conflicts of Interest line
Response: Revised as suggested
- 6, Para. 2, line 284:
This line is unnecessarily scrolled from the previous line, 283. Please correct.
Response: Revised as suggested
Reviewer 3 Report
Comments and Suggestions for Authors
Dear Authors,
Thank you for this interesting article. You can check the format for articles in this journal and introduce the subsections such as material and methods, findings, discussion and then conclusion. In this way, the article will have a seamless flow and link from one section to another.
I would like to express my most sincere gratitude for the opportunity to review this relevant manuscript titled: “Surviving the Storm: The Impact of COVID-19 on Cervical 2 Cancer Screening in Low and Middle-Income Countries.” I have gone through the manuscript and I admit that though it has major weaknesses, the authors can still improve on them to make this work better and publishable. Below are my specific comments.
- In the title, the specificity of the study area is missing. Even if not reflected in the title, the authors should describe the study context in the methodology section.
- The abstract is a stand-alone document. I suggest the authors Write CDC in full.
- I remain cognisant of the word count for this journal. However, the abstract is inadequate as significant components for a stand-alone abstract are missing. For instance, the authors should consider including methods, main findings, and take-home message (s)/conclusion. This section cannot pass in its present form and content.
Introduction:
- In line 35, If the study was conducted in India, then this needs to be reflected in the title of this article.
- In line 50, the authors can consider deleting the word ‘scarce’…how scarce is scarce?
- In line 53, kindly add "was" just after which.
- It is unclear where the authors get this sub-title from “Challenges Imposed by COVID-19 in the Context of Cervical Cancer Screening for Patients, Healthcare Practitioners, and Health Systems in LMIC.” There is a need to signpost where this is originating from.
- In line 61, the authors argue that the study “Presents challenges.” Premised on what methodology?
- The broader objective of this study, as stated in lines 60-61, is not SMART. Which LMCIs did the authors focus on, what duration of the pandemic is the study addressing (is it during the hot-pick or low-pick of the COVID-19 pandemic?)?
- Lines 72-75 need to form part of the methodology.
- Lines 77-78 need a citation before proceeding with the statement.
- The major weakness in this article, which makes it difficult to review the findings and discussion sections, is a lack of methodology section.
- Thus, the authors MUST have a sub-section on methods. This blueprint will inform other readers what was done and how it was done.
- Mentioning 'several approaches' in line 135 sounds like a 'blanket statement". There is a need for specificity by the authors. With such statements, it can be difficult to measure the reliability and validity of this study.
- Line 136, which international studies are the authors referring to? There is a need for citations to authenticate this statement.
Looking forward to reading the revised version of this manuscript.
I wish you all the best.
Reviewer.
Comments on the Quality of English LanguageEnglish is good just a few grammatical errors in the work.
Author Response
|
Response to Reviewer 3 Comments
|
|
|||
|
1. Summary |
|
|
|
|
|
Thank you for dedicating your time to reviewing this manuscript. Below, you will find detailed responses along with corresponding revisions and corrections highlighted in the resubmitted files. Upon thorough examination, it seems that the paper was assessed as if it were an original article. We would like to emphasize that our submission is classified as an opinion piece.
|
|
|||
|
2. Questions for General Evaluation |
Reviewer’s Evaluation |
Response and Revisions |
|
|
|
Does the introduction provide sufficient background and include all relevant references? |
Yes/Can be improved/Must be improved/Not applicable |
|
|
|
|
Are all the cited references relevant to the research? |
Yes/Can be improved/Must be improved/Not applicable |
|||
|
Is the research design appropriate? |
Yes/Can be improved/Must be improved/Not applicable |
|||
|
Are the methods adequately described? |
Yes/Can be improved/Must be improved/Not applicable |
|||
|
Are the results clearly presented? |
Yes/Can be improved/Must be improved/Not applicable |
|||
|
Are the conclusions supported by the results?
|
Yes/Can be improved/Must be improved/Not applicable |
|||
- Point-by-point response to Comments and Suggestions for Authors
Comment 1: Thank you for this interesting article. You can check the format for articles in this journal and introduce the subsections such as material and methods, findings, discussion and then conclusion. In this way, the article will have a seamless flow and link from one section to another.
Response: Thank you for your comment. We would like to mention that we have drafted an opinion piece instead of an original article. Before crafting this paper, we reviewed several samples in the MDPI Healthcare Journal for opinion pieces to develop our approach. Opinions do not have a specific format and can utilize various headings to summarize the evidence and observations.
Comment 2: In the title, the specificity of the study area is missing. Even if not reflected in the title, the authors should describe the study context in the methodology section.
Response: The title does not specify the study area as this opinion piece talks about the impact of COVID-19 on cervical cancer screening in low- and middle-income countries, drawing on our experience from the original study and a literature review. The original paper for this study has been published at: https://bmcpublichealth.biomedcentral.com/articles/10.1186/s12889-023-15602-1.
Comment 3: The abstract is a stand-alone document. I suggest the authors Write CDC in full.
Response: Noted and revised
Comment 4: I remain cognisant of the word count for this journal. However, the abstract is inadequate as significant components for a stand-alone abstract are missing. For instance, the authors should consider including methods, main findings, and take-home message (s)/conclusion. This section cannot pass in its present form and content.
Response: As mentioned earlier, this submission was prepared for the opinion category, where abstracts do not have a structured format.
Comment 5: In line 35, If the study was conducted in India, then this needs to be reflected in the title of this article. In line 50, the authors can consider deleting the word ‘scarce’…how scarce is scarce?
Response: The word scarce has been changed to ‘limited’. As mentioned earlier, the title cannot mention ‘India’ as the opinion piece aims to present the impact of COVID-19 on cervical cancer screening in low- and middle-income countries, drawing on our experience from the original study and a literature review.
Comment 6: In line 53, kindly add "was" just after which.
Response: Noted and revised
Comment 7: It is unclear where the authors get this sub-title from “Challenges Imposed by COVID-19 in the Context of Cervical Cancer Screening for Patients, Healthcare Practitioners, and Health Systems in LMIC.” There is a need to signpost where this is originating from.
Response: As mentioned earlier, this is an opinion piece, affording the authors the liberty to choose sub-headings.
Comment 7: In line 61, the authors argue that the study “Presents challenges.” Premised on what methodology?
Response: The opinion piece aims to present the challenges that COVID-19 has imposed on patients, healthcare practitioners, and health systems regarding cervical cancer screening, along with potential opportunities to address these challenges. It draws on our experience from the original study and a literature review.
Comment 8: The broader objective of this study, as stated in lines 60-61, is not SMART. Which LMCIs did the authors focus on, what duration of the pandemic is the study addressing (is it during the hot-pick or low-pick of the COVID-19 pandemic?)?
Response: The study captured literature from Dec 2019 till the present and all LMICs were included as per World Bank Classification.
Comment 9: Lines 72-75 need to form part of the methodology.
Response: Since it's not an original paper, the journal doesn't specify a particular format for an opinion piece. The description in lines 72-75 simply outlines the approach employed by the opinion paper to organize evidence and insights related to the challenges presented by COVID-19 in cervical cancer screening, with a specific focus on three key stakeholders: patients, healthcare practitioners, and health systems.
Comment 10: Lines 77-78 need a citation before proceeding with the statement.
Response: Thank you for highlighting this. Citations have been added.
Comment 11: The major weakness in this article, which makes it difficult to review the findings and discussion sections, is a lack of methodology section. Thus, the authors MUST have a sub-section on methods. This blueprint will inform other readers what was done and how it was done.
Response: Since it's an opinion piece, the paper does not mention the methodology. The methodology is provided in the original paper: https://bmcpublichealth.biomedcentral.com/articles/10.1186/s12889-023-15602-1
Comment 12: Mentioning 'several approaches' in line 135 sounds like a 'blanket statement". There is a need for specificity by the authors. With such statements, it can be difficult to measure the reliability and validity of this study.
Response: Since it is an opinion paper, it is trying to present evidence on several approaches used to address COVID-19 challenges for cervical cancer screening.
Comment 13: Line 136, which international studies are the authors referring to? There is a need for citations to authenticate this statement.
Response: Thank you for highlighting this. I have included a reference.
Round 2
Reviewer 1 Report
Comments and Suggestions for Authors
Given this is an opinion, it is satisfactory.
Comments on the Quality of English LanguageMinor edits needed.
Author Response
|
Response to Reviewer 1 Comments
|
|||
|
1. Summary |
|
|
|
|
Thank you very much for taking the time to review this manuscript. Please find the detailed responses below and the corresponding revisions/corrections highlighted/in track changes in the re-submitted files.
|
|||
|
2. Questions for General Evaluation |
Reviewer’s Evaluation |
Response and Revisions |
|
|
Does the introduction provide sufficient background and include all relevant references? |
Yes/Can be improved/Must be improved/Not applicable |
|
|
|
Are all the cited references relevant to the research? |
Yes/Can be improved/Must be improved/Not applicable |
|
|
|
Is the research design appropriate? |
Yes/Can be improved/Must be improved/Not applicable |
|
|
|
Are the methods adequately described? |
Yes/Can be improved/Must be improved/Not applicable |
|
|
|
Are the results clearly presented? |
Yes/Can be improved/Must be improved/Not applicable |
|
|
|
Are the conclusions supported by the results? |
Yes/Can be improved/Must be improved/Not applicable |
|
|
|
3. Point-by-point response to Comments and Suggestions for Authors
Comment 1: Given this is an opinion, it is satisfactory. Response: Thank you for accepting our paper as an opinion piece.
Comment 2: Minor edits needed in English language. Response: English language corrections have been made |
|||
Reviewer 3 Report
Comments and Suggestions for Authors
Dear Authors,
Thank you for attending to the reviewer (s) comments which were meant to strengthen and make this article more clearer for a wider readership. Despite the good attempts in attending to the reviewer (s) comments, I have noted the following:
-Despite being an opinion paper, it will be beneficial to have more information on which context (s) you are referring to as Low- and middle-income countries to make it clear whether this is in Africa, Asia, or other countries.
-Thanks for clarifying that this is an opinion paper which does not tie you to a specific format. I also note that despite being an opinion paper, you have now provided additional useful information that strengthens this article for the readers.
-As a way to caution the readers, kindly, include a section on the strengths and weaknesses of this article just before conclusions. This will be mandatory for this kind of article.
I wish you all the best with the other steps in this article.
Thank you.
Reviewer.
Comments on the Quality of English Language
None
Author Response
|
Response to Reviewer 2 Comments
|
|
|||
|
1. Summary |
|
|
|
|
|
Thank you very much for taking the time to review this manuscript. Please find the detailed responses below and the corresponding revisions/corrections highlighted/in track changes in the re-submitted files.
|
|
|||
|
2. Questions for General Evaluation |
Reviewer’s Evaluation |
Response and Revisions |
|
|
|
Does the introduction provide sufficient background and include all relevant references? |
Yes/Can be improved/Must be improved/Not applicable |
|
|
|
|
Are all the cited references relevant to the research? |
Yes/Can be improved/Must be improved/Not applicable |
|||
|
Is the research design appropriate? |
Yes/Can be improved/Must be improved/Not applicable |
|||
|
Are the methods adequately described? |
Yes/Can be improved/Must be improved/Not applicable |
|||
|
Are the results clearly presented? |
Yes/Can be improved/Must be improved/Not applicable |
|||
|
Are the conclusions supported by the results?
|
Yes/Can be improved/Must be improved/Not applicable |
|||
- Point-by-point response to Comments and Suggestions for Authors
Comment 1: Despite being an opinion paper, it will be beneficial to have more information on which context (s) you are referring to as Low- and middle-income countries to make it clear whether this is in Africa, Asia, or other countries.
Response: As advised, we have now indicated this opinion piece is rooted in the Indian context, focusing on the challenges encountered during the PCCIS project; however, the identified challenges and opportunities discussed herein hold broader relevance for other Low- and Middle-Income Countries (LMICs) that lack organized screening programs.
Comment 2: As a way to caution the readers, kindly, include a section on the strengths and weaknesses of this article just before conclusions. This will be mandatory for this kind of article.
Response: Thank you for the comment. As advised, We have now included a section on the strengths and weaknesses of this article just before the conclusion section.
Comment 3: English language editing for the entire paper
Response: As requested, we have reviewed and edited line by line the entire document.